# Melt-Spun Nanocomposite Fibers Reinforced with Aligned Tunicate Nanocrystals

**DOI:** 10.3390/polym11121912

**Published:** 2019-11-20

**Authors:** Alexandre Redondo, Sourav Chatterjee, Pierre Brodard, LaShanda T. J. Korley, Christoph Weder, Ilja Gunkel, Ullrich Steiner

**Affiliations:** 1Adolphe Merkle Institute, University of Fribourg, Chemin des Verdiers 4, 1700 Fribourg, Switzerland; alexandre.redondo@unifr.ch (A.R.); christoph.weder@unifr.ch (C.W.); 2Department of Materials Science and Engineering, University of Delaware, Newark, DE 19716, USA; schatter@udel.edu (S.C.); lkorley@udel.edu (L.T.J.K.); 3College of Engineering and Architecture of Fribourg, University of Applied Sciences of Western Switzerland, Boulevard de Pérolles 80, CH-1705 Fribourg, Switzerland; 4Department of Chemical and Biomolecular Engineering, University of Delaware, Newark, DE 19716, USA

**Keywords:** cellulose nanocrystals (CNCs), polyurethane, nanocomposite fibers, melt-spinning, reinforcement, orientation, thermal stability

## Abstract

The fabrication of nanocomposite films and fibers based on cellulose nanocrystals (P-tCNCs) and a thermoplastic polyurethane (PU) elastomer is reported. High-aspect-ratio P-tCNCs were isolated from tunicates using phosphoric acid hydrolysis, which is a process that affords nanocrystals displaying high thermal stability. Nanocomposites were produced by solvent casting (films) or melt-mixing in a twin-screw extruder and subsequent melt-spinning (fibers). The processing protocols were found to affect the orientation of both PU hard segments and the P-tCNCs within the PU matrix and therefore the mechanical properties. While the films were isotropic, both the polymer matrix and the P-tCNCs proved to be aligned along the fiber direction in the fibers, as shown using SAXS/WAXS, angle-dependent Raman spectroscopy, and birefringence analysis. Tensile tests reveal that fibers and films, at similar P-tCNC contents, display Young’s moduli and strain-at-break that are within the same order of magnitude, but the stress-at-break was found to be ten-times higher for fibers, conferring them a superior toughness over films.

## 1. Introduction

Driven by environmental and ecological concerns, biopolymers are explored as an alternative to conventional, petrochemical-based polymers. Cellulose, the most abundant biopolymer on earth, has therefore attracted much attention [1]. Cellulose is a polymer that is bio-synthetized by plants, bacteria, and tunicates–invertebrate marine animals. Native cellulosic materials can be degraded into cellulose nanocrystals (CNCs) using acid hydrolysis. These rod-like nanoparticles exhibit diameters and lengths ranging from 5–30 nm and 50–3000 nm, respectively, depending on the biological source and the isolation process [2]. Because of their low density, high stiffness, and high aspect ratio, CNCs have been widely utilized as nano-fillers for the reinforcement of polymers [1,3]. They are further of interest because of their biodegradability, renewability, recyclability, and biocompatibility, and they pose considerably lower health risks than other nanofillers [2,4,5]. The extraction and isolation of CNCs have matured to the point that industrial procedures have been established, and therefore CNCs can already be obtained in large quantities at a low cost, making them readily available to be used at an industrial scale [5]. Among the different sources of celluloses, tunicates have been shown to afford CNCs with the highest aspect ratio, and therefore CNCs isolated from tunicates (tCNCs) confer better mechanical reinforcement properties to polymer matrices than CNCs obtained from other sources [6,7].

Within the large group of CNC-reinforced polymers [8], many studies focused on nanocomposites made of polyurethanes (PUs) and CNCs [9,10,11,12,13,14,15,16,17,18,19,20,21]. Besides their capability of being melt-processable, PUs are appreciated for offering high strength, good elasticity, and molecular tenability [22,23,24]. Most of the PUs used were microphase separated thermoplastic elastomers that consist of a rubbery phase formed by a non-crystallizing telechelic having a low glass transition temperature (*T_g_*) and hard segments composed of diisocyanate and diols that serve as physical cross-links [10]. PU/CNC nanocomposites are typically processed into films by solvent casting [11,15,17,25] or by melt-mixing and subsequent compression molding [17]. For example, Natterodt et al. demonstrated that solvent-cast PU/CNC nanocomposite films with a CNC content of 30 wt% considerably improve the strength and stiffness of the polymer matrix while maintaining a reasonable strain-at-break [17]. Similarly, Nicharat et al. used a melt-mixing process to produce PU/CNC nanocomposites that display much higher stiffness than neat PU without overly compromising the extensibility [20]. The increase in stiffness achieved upon incorporation of CNCs into polymer matrices is usually counterbalanced by a significantly reduced elongation at break, reducing the material’s toughness. However, such embrittlement is typically not observed when PUs are employed as matrices. Previous studies with nanoclays suggest that this is associated with the preferred incorporation of the CNCs in the hard segments of PU, and perhaps because stress-transfer between the CNCs occurs through the rubbery PU phase [26].

Generally, the enhancement in stiffness and strength can be further increased if the matrix polymer chains and the CNCs are uniaxially oriented, as previously demonstrated using poly(vinyl acetate) [27], and poly(vinyl alcohol) [28] matrices, as well as neat CNC films [29,30,31,32]. Different strategies have been explored to control the orientation of the CNCs within various polymer matrices, such as shear casting films [29], magnetic field-induced alignment of CNCs [30], or by processing nanocomposites into fibers [31,32,33,34,35,36,37,38]. Reising et al. demonstrated the importance of controlling the CNC orientation to achieve superior mechanical properties compared to isotropic systems [29]. Neat CNC films were solvent-cast and shear-cast to induce alignment of the CNCs along the shear direction. Mechanical properties were measured along the shear direction for shear-solvent cast films and solvent cast films. When elongated in the shear direction, films that were shear-cast from a solvent exhibited an increase in the Young’s modulus by a factor of two compared to films produced by solvent casting alone [29]. Nanocomposite fibers containing CNCs have been produced by different techniques such as wet-spinning [33,39,40], dry-spinning [31,37], electrospinning [32,41], or melt spinning [34,35,36,42]. Amongst these spinning methods, melt spinning is of particular interest since it is not only a solvent-free process, and thus easily scalable, it also efficiently orients the polymer chains along the drawing direction because of high shear forces. However, melt processing of CNC-containing composites is significantly limited by the poor thermal stability of CNCs extracted by sulfuric acid-based hydrolysis. During this hydrolysis, sulfate groups are grafted and adsorbed to the surface of the CNCs, increasing their electrostatic repulsion but catalyzing their thermal degradation [8,20]. Therefore, melt spinning of CNC-based fibers so far has been limited to polymer matrices with comparably low processing temperatures, e.g., acetate butyrate [36], and polylactic acid [43], and often required an inert atmosphere. Despite comparably high processing temperatures, melt-spun fibers made of PU and CNCs isolated by sulfuric acid hydrolysis were recently demonstrated [38]. Interestingly, the PU was found to be isotropic in both fibers and solvent-cast films, whereas CNCs were isotropic in films, but aligned in fibers. However, the alignment of the CNCs was not found to additionally contribute to the mechanical reinforcement of the nanocomposite. Therefore, effective molecular alignment of *both* constituents as well as the effect of alignment on the mechanical properties of PU/CNC fibers have yet to be demonstrated.

Here, we report the melt-spinning of thermally stable PU/CNC fibers and show that mechanical reinforcement can be improved, while a high degree of elasticity is maintained, by controlling the orientation of both the CNCs and the PU chains. In order to maximize the mechanical reinforcement of the nanocomposite, and ensure thermal stability of the CNCs, high-aspect-ratio CNCs were isolated from tunicates, using phosphoric acid hydrolysis [9,20,44]. Nanocomposite films and fibers were produced by solvent casting or melt-mixing in a twin-screw extruder and subsequent melt-spinning, respectively. We demonstrate the alignment of both CNCs and PU in fibers, while in comparison films remain entirely isotropic as determined from structural analyses using small- and wide-angle X-ray scattering, qualitative birefringence analysis, and angle-dependent Raman spectroscopy. Moreover, mechanical analysis of films and fibers reveals that molecular alignment significantly contributes to the superior mechanical properties of fibers compared to films.

## 2. Materials and methods

### 2.1. Materials

Texin 985, a thermoplastic polyurethane (PU) based on poly(tetramethylene glycol), butanediol, and 4,40-methylenebis(phenyl isocyanate) with a Shore A hardness of about 85 was obtained from Covestro [45]. Dimethylformamide (DMF), sulfuric acid 95–98%, and phosphoric acid 85% were obtained from Sigma-Aldrich Corp. (Saint-Louis, MO, USA) and were used as received without further purification.

### 2.2. Isolation of Cellulose Nanocrystals (P-tCNCs) Using Phosphoric Acid Hydrolysis

Tunicates collected from the French Atlantic coast were purified using a Soxhlet extraction and a subsequent bleaching step following procedures described elsewhere [46]. The extracted material was dried and ground into a powder using a Blender Philips HR2195/00 (PHILIPS, Amsterdam, Netherlands) at maximum speed (900 W output power). The hydrolysis conditions used here represent an adaptation of the procedure described by Camarero Espinosa et al. [44] for cotton CNCs. A 75 wt% phosphoric acid solution was prepared by adding 412 mL of 85 wt% phosphoric acid to 88 mL of deionized water, under vigorous stirring. The acid solution was heated to 100 °C, before 2 g of the ground, dried, and bleached tunicate mantles were added, and the mixture was stirred for 4 h at 100 °C. The mixture was then cooled to 25 °C by the addition of 500 g of ice and centrifuged at 14,500 RPM for 30 min. The supernatant was decanted, replaced with deionized water, and the centrifugation step was repeated at least two times until the supernatant was transparent and colorless. The remaining suspension was then dialyzed against deionized water for five days, exchanging the water every day. The final P-tCNC suspension, having a pH of about 5.5–6, was redispersed by sonication for 15 min using a horn ultrasonicator (Branson Digital Sonifier S-250D, Branson Ultrasonics, Danbury, CT, USA, 50−60 Hz/ 200 W) at 15% amplitude. The dispersed suspension was frozen overnight in a conventional freezer and lyophilized in a lyophilizer (Telstar LyoQuest Laboratory Freeze Dryer, Terassa, Spain) for three days. The procedure afforded P-tCNCs as a white cotton-like material in a yield of 55–65%.

### 2.3. Isolation of Cellulose Nanocrystals (S-tCNCs) Using Sulfuric Acid Hydrolysis

Tunicate cellulose was hydrolyzed by dispersing 6 g of ground, dried, and bleached tunicate mantles in 260 mL of water and subsequently slowly adding 280 mL of sulfuric acid (95–98%), maintaining a temperature below 30 °C using an ice bath. After the addition was complete, the mixture was heated to 45 °C and stirred at this temperature for 4.5 h. The mixture was then cooled to room temperature by the addition of 500 g of ice, and the workup mirrored the above protocol. The procedure afforded S-tCNCs as a white cotton-like material in a yield of 50%.

### 2.4. Nanocomposite Fibers Melt Spinning

Nanocomposite fibers with a P-tCNC content of 1 wt% or 5 wt% were prepared using a DACA microcompounder (DACA Instruments, Santa Barbara, CA, USA). To ensure a good dispersion of the P-tCNCs in the PU matrix, 40 mg or 200 mg of P-tCNCs were first dispersed in 20 mL of DMF by sonication in an ultrasonic bath for 1 h. Total of 8 g of PU was separately dissolved into 100 mL of DMF to create a solution with a polymer concentration of 80 mg/mL. Aliquots of 5 or 25 mL of the PU solution were combined with 20 mL of DMF containing 40 mg and 200 mg of P-tCNC dispersions, respectively, to create mixtures containing 10% w/w P-tCNCs with respect to total solid content. The solutions were stirred overnight and cast into PTFE Petri dishes having a diameter of 120 mm. The solvent was evaporated by initial oven-drying at 50 °C for two days, and additional drying in a vacuum oven at 50 °C for one day. To obtain a final P-tCNC concentration of 5 wt% and 1 wt% in nanocomposite fibers, films containing 200 mg and 40 mg of P-tCNCs, respectively, prepared as described above, were fed into a DACA MicroCompounder with 2 g and 3.6 g of neat PU pellets. These materials were then mixed between co-rotating screws for 5 min at 210 °C and 50 RPM. The melt streams were then extruded through a die having a diameter of 0.5 mm and collected on a take-up wheel, which was located at a distance of 15 cm from the extruder, rotating at a speed of 22 m/min. Neat PU fibers were produced by the same procedure, using neat PU pellets only.

### 2.5. Nanocomposite Film Solvent Casting

Nanocomposite films with a P-tCNC content of 1 wt% and 5 wt% were prepared by solvent casting. 1 mg and 5 mg of P-tCNCs were dispersed into 10 mL of DMF by sonication for 1 h in an ultrasonic bath. 2 × 100 mg of PU were dissolved in 2 × 10 mL of DMF. The P-tCNC dispersions and PU solutions were combined and stirred overnight to create mixtures with 1 wt% and 5 wt% P-tCNCs with respect to the total solid content. These were cast into 7 cm diameter PTFE Petri dishes, oven-dried at 50 °C for three days followed by drying in a vacuum oven at 50 °C for one day. A film thickness of around 30 micrometers was measured for films of both compositions using a Mitutoyo MDH high-accuracy sub-micron digital micrometer. Neat PU films were produced by the same procedure, using neat PU pellets only.

### 2.6. Methods for Characterization

Scanning Electron Microscopy (SEM): Nanocomposite fibers were imaged using a Tescan Mira 3 LMH scanning electron microscope at 4 kV (Tescan, Brno, Czech Republic). Samples were prepared by depositing 3 nm of a conductive layer of gold using a 208 HR Cressington sputter coater. The images were processed and analyzed using ImageJ software.

Thermogravimetric Analysis (TGA): A Mettler-Toledo STAR thermogravimetric analyzer system (Mettler-Toledo, Greifensee, Switzerland) equipped with Al_2_O_3_ crucibles was used for thermogravimetric analysis in a temperature range of 0 to 600 °C at a heating rate of 10 °C/min under air atmosphere.

Atomic Force Microscopy (AFM): AFM imaging of CNCs was performed with a Nano Wizard II (JPK BioAFM, Berlin, Germany) microscope operated in tapping mode using aluminum-coated silicon probes (NanoAndMore) with a nominal force constant of 40 N/m, a nominal resonance frequency of 300 kHz, and a tip radius <10 nm. Samples were prepared by depositing 40 μL of an aqueous CNC solution (0.05 mg/mL) onto freshly cleaved mica and subsequent drying overnight at room temperature. AFM images were processed and analyzed with Gwyddion and ImageJ software. Based on three AFM images, 129 individual P-tCNCs were analyzed, and their lengths were measured.

Polarized Optical Microscopy (POM): An Olympus BX51 microscope was used in transmission mode for all experiments. The birefringence was monitored between linear crossed polarizers (Olympus U-POT) and an Olympus DPT2 camera (Olympus Schweiz AG, Volketswil, Switzerland).

Tensile Testing: Tensile tests of nanocomposite fibers and films were carried out using a Zwick/Roell Z010 10KN material tensile testing machine (Zwick Roell, Ulm, Germany). The instrument was equipped with a 200 N load cell and a 2.5 kN clamp. Fibers were directly mounted, and films were cut into rectangles of 50 mm × 5 mm with a thickness ranging from 30 to 40 micrometers. Samples were tested with a strain-rate of 50 mm/min and a pre-load force of 0.005 N. Cyclic tensile tests were performed under similar conditions at 15%, 30%, 45%, 60%, 75%, 90%, and 105% strain.

Small-angle X-ray Scattering and Wide-angle X-ray Scattering (SAXS and WAXS): SAXS and WAXS measurements of nanocomposite samples were performed at the University of Delaware on a Xeuss 2.0 SAXS/WAXS system (Xenocs, Sassenage, France) operated at a power of 50 kV/0.6 mA. Monochromatic X-rays having wavelength of 1.542 Å (Cu Kα radiation) were used to irradiate samples at a sample-to-detector distance of 72 mm for WAXS and 1200 mm for SAXS measurements, determined by calibration using a silver behenate standard. Exposure times were 1 h, and scattering patterns were recorded on a Pilatus detector (comprised of three panels) having a total number of pixels of 486 × 618.

Raman Microscopy: Raman scattering was excited using a HeNe laser at 633 nm, and spectra were recorded using a LabRAM HR800 confocal micro-Raman spectrometer (HORIBA, Kyoto, Japan) equipped with an Olympus BX41 optical microscope, a high-resolution diffraction grating (1800 gr/mm), and a Peltier-cooled CCD detector. The laser was focused onto the fibers and films by a 100× objective, providing a laser spot size of 1 μm. The spectra were collected between 600 cm^−1^ and 4000 cm^−1^ using an average acquisition time of five seconds. Single fiber strands were mounted on a rotatory stage that allows the azimuthal rotation of the sample. Spectra were recorded at 20° intervals for azimuthal angles between 0° to 360°.

Dynamic Scanning Calorimetry (DSC): Differential scanning calorimetry measurements were performed using a Mettler-Toledo STAR system ((Mettler-Toledo, Greifensee, Switzerland) under N_2_ atmosphere. The procedure used included two heating and cooling cycles of an approximately 6 mg sample in a 40 µl closed aluminum pan from −80 °C to 180 °C at a heating rate of 10 °C/min.

Infrared Spectroscopy (IR): Spectra were recorded on a Perkin Elmer Spectrum 65 FTIR spectrometer (Perkin Elmer, Waltham, MA, USA) between 4000 and 600 cm^−1^ at a resolution of 4 cm^−1^, an acquisition time of 5 s, averaging five consecutively collected spectra.

Nuclear Magnetic Resonance (NMR): ^1^H-NMR spectra were acquired in deuterated DMF on a Bruker AVANCE III 400 spectrometer (Bruker, Billerica, MA, USA) operated at 400 MHz. The spectra were analyzed and processed using the MestreNova software (Mestrelab Research, Santiago de Compostela, Spain).

## 3. Results and Discussion

### 3.1. P-tCNC Extraction and Characterization

CNCs can be extracted and isolated from many different natural sources. Depending on the hydrolysis conditions and cellulose sources, the CNC crystallinity and aspect ratio differ greatly [2]. High-aspect-ratio CNCs, extracted from tunicates, reinforce polymer matrices more effectively than low-aspect-ratio CNCs, extracted from wood or cotton [2,20,47]. While the cellulose source determines the aspect ratio of the CNCs, the extraction procedure of the crystals determines their thermal stability. For example, sulfuric acid hydrolysis affords rather low degradation temperatures, whereas phosphoric acid hydrolysis provides high enough thermal stability for melt processing because of a lower amount of degradation-catalyzing charged groups at the CNC surface, as discussed in detail by Espinosa et al. [12] for cotton CNCs, and by Nicharat et al. for tunicate CNCs [20,44]. To achieve high-aspect-ratio CNCs with high thermal stability, CNCs extracted from tunicate cellulose by phosphoric acid hydrolysis were used in this study. The quantitative analysis of atomic force microscopy images (Figure 1a,b) shows the high aspect ratio of these CNCs, with an average length of 1.61 ± 0.76 µm, in agreement with the literature [7,48,49,50,51]. The high thermal stability of P-tCNCs was confirmed by TGA measurements (Figure 1c) showing an onset degradation temperature of about 300 °C, which is significantly higher than that of tunicate CNCs extracted using sulfuric acid hydrolysis (~180 °C). Previously, and in agreement with the above-mentioned result, Nicharat et al. reported an onset degradation temperature of 180 °C for S-tCNCs and 290 °C for P-tCNCs [20]. Note that the high onset degradation temperature of P-tCNCs enables the melt extrusion of PU/P-tCNC nanocomposites at around 210 °C, which is not the case for S-tCNCs, where CNC degradation is expected [45].

### 3.2. Fiber Processing

The efficient dispersion of CNCs in a polymeric matrix is essential for achieving strong mechanical reinforcement. To this end, both effective matrix-CNC and CNC-CNC interactions are a prerequisite, enabling an effective fiber reinforcement of the polymer matrix. Given the hydrophilic nature of CNCs because of the presence of OH groups on their surface, efficient CNC-polymer interactions require a matrix polymer containing hydrogen-bond acceptor and donor functional groups. Unless specific processing protocols [52,53,54] or dispersing agents [55] are used, the incorporation of CNCs into less polar matrices results in limited mechanical reinforcement, because of their aggregation and lack of stress-transfer from the polymer matrix to the CNCs. However, the PU contains rather polar urethane groups that enable hydrogen-bonding with the OH groups on the CNC surface as confirmed by IR and NMR measurements (Appendix A). The fact that previous studies have shown that the level of reinforcement upon introducing CNCs into PUs followed the predictions of a percolation model suggests that indeed good dispersion is possible [10]. To maximize the dispersion of the P-tCNCs, the PU used here is a commercial thermoplastic elastomer comprised of poly(tetramethylene glycol), butanediol, and 4,4′-methylenebis(phenyl isocyanate) with a Shore A hardness of about 85. The molecular composition renders this PU favorable for the stabilization of individual P-tCNCs through hydrogen bonding and therefore enables efficient dispersion of the P-tCNCs. A solvent-based pre-dispersion step was applied, in which a solution of the P-tCNCs in DMF was combined with a mixture of the polymer before the solvent was evaporated. These “masterbatches” had P-tCNC contents of 10 wt%. They were diluted by mixing with neat PU to form nanocomposites containing 0, 1, or 5 wt% of P-tCNCs. Low P-tCNCs content melt-spun fibers are predicted to lie below the CNC percolation threshold (~7 wt%), as previously determined by Mendez et al. for Texin 985/CNC nanocomposites [10]. We hypothesize that sub-percolation concentration nanocomposites are easier to orient than materials having a CNC content above this limit.

Melt-spinning was carried out with a co-rotating twin-screw microextruder, which was used to pump the melt through a spinneret featuring a single round hole (Figure 2). The monofilaments were solidified upon cooling in air before collection on a take-up wheel. This process imparted uniaxial deformation of the polymer chains and the P-tCNCs along the fiber axis that was retained in the solid state. Increasing the amount of P-tCNCs in PU-based nanocomposite fibers was not found to change their thermal stability at the concentrations employed here, as confirmed by TGA (Appendix A). Figure 3 shows that the viscosity increase associated with the incorporation of CNCs into the PU, similar to previous reports [35], led to an increase of the fiber diameter from ca. 68 ± 15 µm (neat PU) to 136 ± 10 µm (5 wt% CNCs) as well as an increase in fiber roughness (the fiber diameters were measured by a micrometer). Note that such a thickness increase is known to affect the mechanical properties of PU fibers, decreasing the stress-at-break while increasing the strain-at-break [56]. In addition, increasing the P-tCNC content in the composite materials embrittles the material, rendering the fibers more prone to cracking or breaking under processing conditions where the neat PU fibers remain intact. 

### 3.3. Mechanical Properties

The uniaxial deformation experienced by the fibers during melt-spinning in both the molten and in the solid state affords nanocomposite fibers in which the polymer chains and CNCs adopt a preferred orientation along the fiber direction. Unidirectional molecular and CNC orientation are well-known to lead to an increase in the tenacity and the toughness along the direction of orientation compared to isotropic nanocomposites [28,29,31,32,57]. To examine the effect of molecular orientation on the mechanical properties of nanocomposites, PU/P-tCNC melt-spun fibers, and solvent-cast PU/P-tCNC nanocomposite films were compared. Note that while slight differences in thermal history between nanocomposite fibers and films were observed in DSC measurements (Appendix A), such differences were previously shown to have no impact on their mechanical properties [38]. The mechanical properties of fibers and films were then evaluated using the stress–strain measurements with representative curves shown in Figure 4. The corresponding tensile characteristics such as Young’s modulus, strain-at-break, stress-at-break, and toughness (as determined from integrating of the stress-strain curves) are summarized in Table 1.

A comparison of the mechanical properties of the neat PU films and fibers shows that the stiffness of fibers increases by a factor of 2.7 compared to that of films, while the tenacity is enhanced by a factor 14. In both films and fibers, the incorporation of CNCs led to a significant (6–13-fold) increase in the Young’s modulus, while the maximum strain was reduced to about 50% of the value of neat samples upon the introduction of 5% P-tCNCs. Interestingly, the extent of reinforcement is similar for films and fibers, the Young’s modulus increased from 7 (neat PU) to 100 MPa (5 wt% P-tCNCs) in the films, whereas an increase from 20 to 123 MPa is observed in the fibers. This agrees with the previous work on similar PU/CNC nanocomposites, where the extent of CNC-reinforcement was similar for films and melt-spun fibers [38]. In contrast, the CNCs impact the stress-at-break differently in fibers and films. Upon increasing the CNC content, the stress-at-break increased in films, but decreased in fibers. Despite these differences, the nanocomposite fibers exhibit 6 to 14 times higher stress-at-break values compared to films, within the CNC concentration range of this study. As discussed in more detail below, the molecular orientation of the PU chains and CNCs is very different in fibers and films, accounting for the observed differences in the mechanical properties. Despite the apparent reduction of the strain-at-break upon P-tCNC addition, both films and fibers retained extensibilities of above 200% for nanocomposites containing 5 wt% P-tCNCs. In fibers, this reasonable extensibility is combined with a high stress-at-break, yielding a maximum toughness of up to 2.8 MJ/m^3^, which exceeds that of the corresponding composite film by one order of magnitude (0.3 MJ/m^3^).

To gain a better understanding of how the microscopic structure affects the mechanical properties of the nanocomposites, cyclic strain-stress measurements were performed on 5 wt% P-tCNC nanocomposite fibers, as shown in Figure 5.

The cyclic test in Figure 5a shows the progressive softening of the nanocomposites, which leads to a decrease in Young’s modulus as the maximum strain of the cycle is increased (Figure 5b). In an attempt to recover the initial mechanical properties, the fibers were thermally annealed at 50 °C for three days. The full recovery of the initial Young’s modulus was, however, not achieved. Nonetheless, some recovery was observed, amounting to a factor of two in Young’s modulus compared to the value prior to thermal annealing. It seems that the softening of the nanocomposites only occurs for an elongation lower or equal to the maximum strain previously applied to the sample. If, however, the extension reaches a strain higher than the previously applied maximum strain, the stress–strain response matches that of a single uniaxial tensile test (Figure 5a, black curve) [58]. This effect has already been observed in rubber and pure crystalline gums by Mullins et al. in 1969 and, given the structural complexity of the materials that exhibit this effect, its microscopic origin is still not sufficiently explained [59].

### 3.4. Orientation of Matrix Polymers and CNCs in Nanocomposite Films and Fibers

The PU/CNC nanocomposite fibers investigated here include two ordered components, the PU hard domains, and the embedded CNCs. Characterizing the orientation of both, the CNCs and PU hard segments with respect to the macroscopic fiber axis is of fundamental interest, since orientation in fibrous materials is intimately coupled to their macroscopic mechanical properties. The alignment of PU hard segments and P-tCNCs was studied by Raman spectroscopy and wide-angle X-ray scattering (WAXS). This alignment induces also an angle-dependent birefringence in the nanocomposite fibers, which was visualized by polarized optical microscopy.

Images that showcase angular birefringence measurements of fibers are shown in Figure 6a. The fiber birefringence varies strongly with their relative position between crossed polarizers. When the fibers are parallel to one of the two polarizers, extinction of the birefringence is observed. Upon the rotation of the sample, a progressive increase of birefringence is detected, with a maximum at 45°. This angle-dependent behavior in nanocomposite fibers demonstrates their overall highly anisotropic nature, with a preferred orientation along the fiber axis. A similar experiment was carried out with a nanocomposite film containing 5 wt% P-tCNCs, as shown in Figure 6b, where no variation of the birefringence intensity with film rotation between crossed polarizers was observed, confirming the isotropic nature of the nanocomposite films.

This angular birefringence measurement enables a fast and straightforward qualitative assessment of the orientation of ordered domains within materials. The birefringence contribution arises from the crystalline nature of the CNCs and the PU hard segments. Because P-tCNCs are present at very low concentrations, further characterization was conducted to demonstrate their orientation within the matrix. Raman spectroscopy is known to be a powerful tool for the determination of the orientation of crystalline structures. It has been employed for the evaluation of the alignment of carbon nanotubes [60,61], crystalline polymers [62,63,64], or cellulose nanocrystals [65,66,67,68]. The most intense band in the cellulose spectrum is located at 1095 cm^−1^, arising from the vibrations of the C–C and C–O bonds of the polymer backbone [10,33,69]. This signal was used to investigate the orientation of the P-tCNCs in nanocomposite fibers and films. The molecular orientation of PU was also investigated by angle-dependent Raman spectroscopy. The Raman band at 1615 cm^−1^, corresponding to the C–C ring stretching mode of the benzene rings of the diphenylmethane motif, was used to characterize the PU orientation. Raman spectra of the nanocomposite fibers containing 5 wt% P-tCNCs are shown in Figure 7a. Only two P-tCNC Raman bands are observable; the one of interest at 1095 cm^−1^, which was used for the evaluation of the orientation, and another band at 3500 cm^−1^ corresponding to the O–H vibration.

The angular dependences of the two characteristic Raman bands associated with the PU matrix and the CNCs in the 5 wt% nanocomposite fibers are shown in Figure 7b. The graphs reflect that the signal intensities are indeed highly dependent on the sample orientation, with intensity maxima at 90° and 270°, i.e., in arrangements where the fiber direction is parallel to the polarization of the laser. Intensity minima were detected when fibers are oriented perpendicular to the laser polarization, at 0° and 180°. These results suggest that the PU hard segments and P-tCNCs are predominantly aligned along the fiber axes. Nanocomposite films were also tested, and their angular Raman plots show no preferential orientation of either the PU hard domains or the P-tCNCs, confirming the isotropic nature of the film, as shown in Figure 7c. Note that the processing-induced PU alignment in fibers observed here was not seen in previously reported melt-spun PU/CNC nanocomposite fibers [38].

While Raman spectroscopy is a powerful tool to evaluate the degree of alignment, it is a surface-sensitive technique since the laser beam penetrates only a few micrometers into the sample. To further elucidate the orientation of the P-tCNCs and PU in the bulk of the nanocomposites, SAXS and WAXS characterizations were conducted. 2D WAXS and SAXS spectra are presented in Figure 8, providing information concerning the orientation of the PU hard segment and P-tCNCs within the matrix. No reflections arising from CNCs are present in the SAXS spectra, and PU orientation can therefore be evaluated directly from SAXS spectra, shown in Figure 8a,c. No particular reflection features can be observed in 2D SAXS and 2D WAXS spectra of films, confirming their isotropic nature. On the other hand, the SAXS pattern of the nanocomposite fibers exhibits a two-point pattern, suggesting an orientation of the P-tCNCs along the fiber axes, as previously reported [70,71,72,73]. Figure 8b shows 2D WAXS spectra of fibers, which give insight into P-tCNC orientation. Whereas an isotropic ring is observed for P-tCNC reflections in the WAXS pattern of films, an arc-like pattern is observed in the WAXS images of P-tCNC nanocomposite fibers. To gain better insight into the anisotropy of P-tCNCs, the azimuthal integration over 360° at *q* = 1.21 ± 0.01 Å^−1^ is shown in Figure 8e. P-tCNC reflections from the nanocomposite fibers show pronounced peaks in intensity at 90° and 270° demonstrating the anisotropic nature of the materials. In contrast, P-tCNC reflections from films form a full ring pattern lacking any significant angular variation in the intensity (Figure 8e).

Azimuthal integration of the 2D WAXS images provides information on the characteristic reflections of crystals within the nanocomposite materials. Figure 9 shows that neat PU fibers and films possess the same intense reflection centered at 1.1 Å^−1^. Interestingly, films exhibit another reflection located at 1.17 Å^−1^, which is not detected in the fiber spectrum. This additional reflection might be due to the difference in processing. During solvent casting, the slower evaporation of DMF may promote self-assembly in the nanocomposite, increasing order within the material. Characteristic P-tCNC reflections at 0.8 Å^−1^, 0.9 Å^−1^, 1.24 Å^−1^, and 1.92 Å^−1^ are also shown and indexed in Figure 9. Other CNC reflections might be obscured by the intense PU signal.

## 4. Conclusions

We demonstrate that the incorporation of thermally stable CNCs within a matrix of PU significantly improves the stiffness of the nanocomposites. By preparing CNCs via phosphoric acid hydrolysis, the resulting P-tCNCs were suitable for melt spinning at 210 °C. Increasing the P-tCNC content from 0 to 5 wt% significantly increased the stiffness of the composite while retaining a reasonable extensibility. The reinforcement was found to be more effective in melt-spun fibers compared to solvent-cast films. More specifically, P-tCNC-reinforced fibers exhibited a ten-fold increased toughness compared to the films. Raman spectroscopy and SAXS/WAXS analysis showed a strong alignment of both the P-tCNCs and the PU hard domains in the direction of the fiber axis, which is contrast to their isotropic orientation in films. Interestingly, for sufficiently high P-tCNC loading, the increase in elastic modulus is similar for the fibers and films. The actual benefit of alignment lies in the stress-at-break (and thus the toughness), which is higher by one order of magnitude in fibers compared to films, with a relatively low (~200% for 5% CNC loading) embrittlement upon CNC addition, i.e., at comparably low P-tCNC concentration. The PU alignment is the major contribution to the nanocomposite strength while P-tCNCs stiffen the material irrespective of their orientation.

Further investigations should address the integrity of P-tCNCs in melt-extruded fibers, the nature of their bonding to the PU matrix, and the variation of the mechanical properties of the composite with fiber diameter.

## Figures and Tables

**Figure 1 polymers-11-01912-f001:**
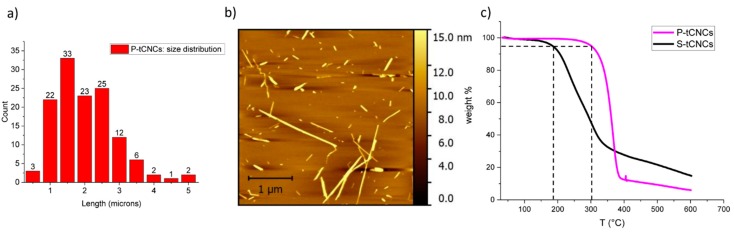
(**a**) P-tCNC size distribution, determined from atomic force microscopy (AFM) images, as the one shown in (**b**). (**c**) Thermogravimetric analysis (TGA) traces of cellulose nanocrystals (CNCs) extracted from tunicates cellulose with phosphoric acid (P-tCNCs, red) and sulfuric acid (S-tCNCs, black), showing the superior thermal stability of P-tCNCs with a degradation onset of 300 °C compared to 180 °C for S-tCNCs.

**Figure 2 polymers-11-01912-f002:**
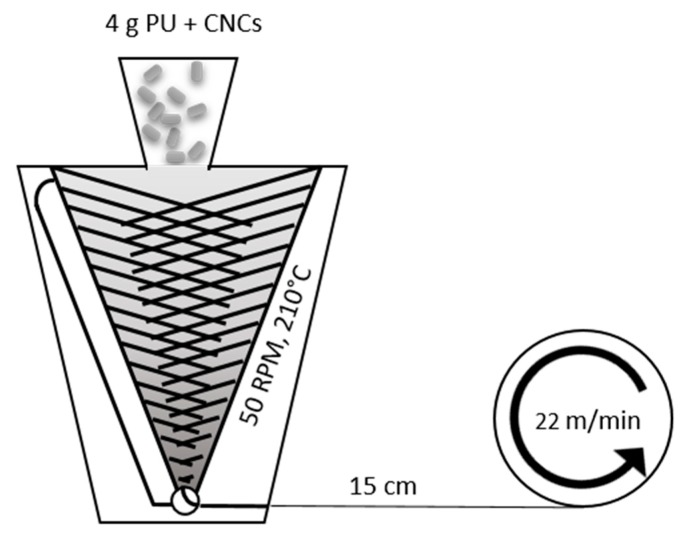
Schematic representation of the melt-spinning set-up.

**Figure 3 polymers-11-01912-f003:**
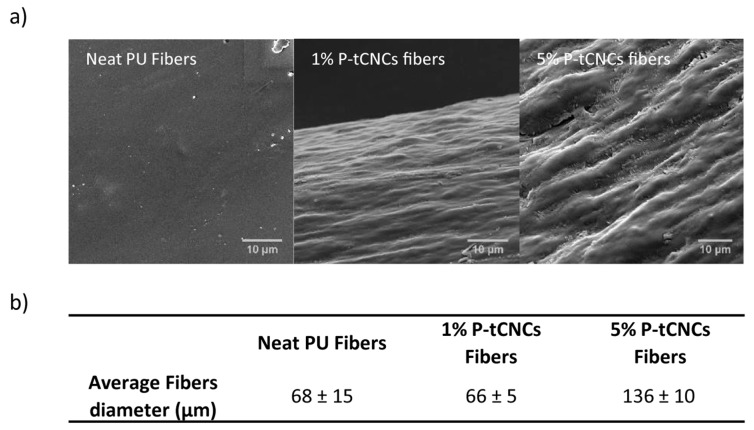
(**a**) Scanning electron microscopy (SEM) images of a neat polyurethane (PU) fiber, and fibers containing 1 or 5 wt% P-tCNCs. The increase in CNC content leads to significant surface roughness and surface defects. (**b**) Average diameters of fibers as determined by micrometer measurements.

**Figure 4 polymers-11-01912-f004:**
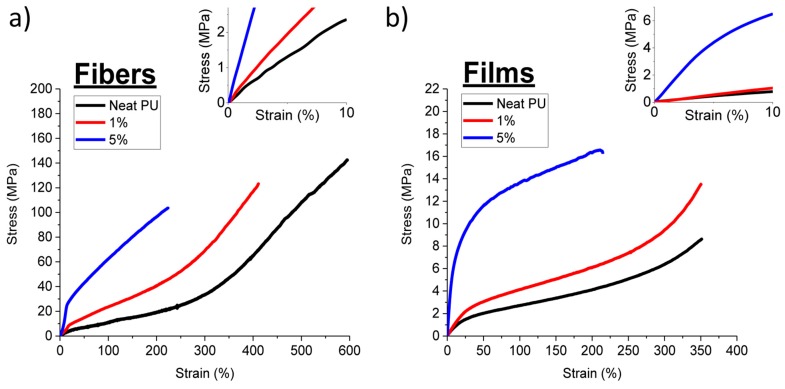
Representative stress-strain curves of (**a**) P-tCNC nanocomposite fibers and (**b**) P-tCNC nanocomposite films. The insets are magnifications near the origin of the graphs, which were used to determine the Young’s modulus of the materials.

**Figure 5 polymers-11-01912-f005:**
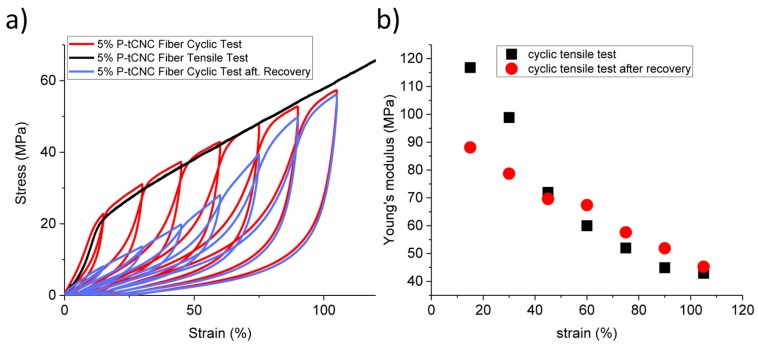
(**a**) Cyclic tensile test experiments performed on 5 wt% P-tCNC nanocomposite fibers, raising the strain to 5%, 15%, 30%, 45%, 60%, 75%, 90%, 105%, followed by unloading the sample (red). After a full series of cycles, the fibers were annealed for 3 days at 50 °C and the measurement was repeated (blue). A simple non-cyclic tensile test is shown for comparison (black). (**b**) Evolution of Young’s modulus during the cyclic tensile test before and after recovery, showing that the initial stiffness can only be partially recovered.

**Figure 6 polymers-11-01912-f006:**
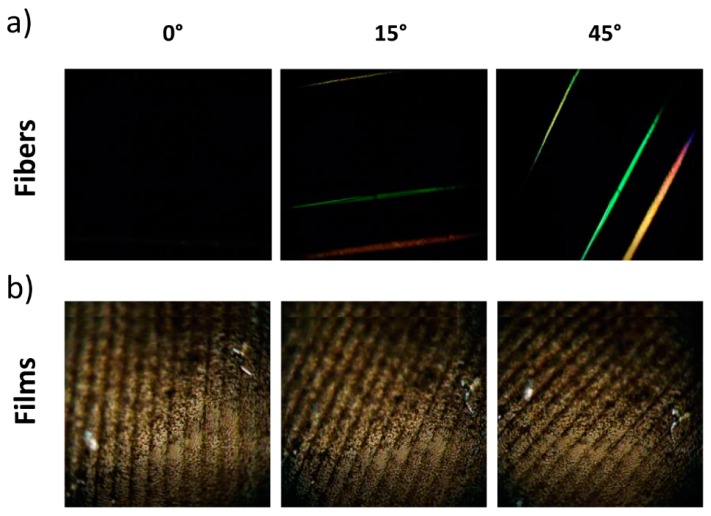
(**a**) 5 wt% P-tCNC fiber (top), neat PU fiber (middle), 1 wt% P-tCNC fiber (bottom) between crossed polarizers oriented at 0°, 15°, 45° relative to the optical filters, showing the anisotropic nature of the fibers. (**b**) 5 wt% P-tCNC nanocomposite film between crossed polarizers oriented at 0°, 15°, 45° relative to the optical filters, showing the isotropic nature of nanocomposite films.

**Figure 7 polymers-11-01912-f007:**
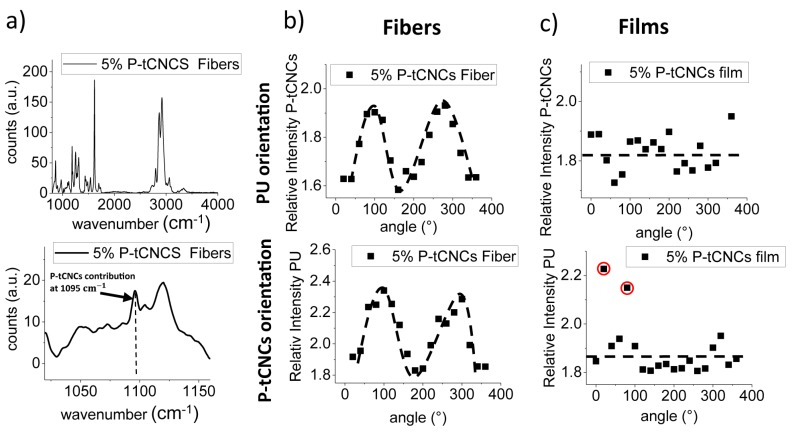
(**a**) Raman spectra of 5% P-tCNC containing nanocomposite fibers with the inset showing a zoom into the region of interest, highlighting the CNC contribution at 1095 cm^−1^. (**b**) 1D angular Raman plot for 5% P-tCNC nanocomposite fibers, showing the strong angular dependence of the 1095 cm^−1^ Raman signal. (**c**) 1D angular Raman plot for 5% P-tCNC nanocomposite films, showing a lack of angular dependence of the Raman signal. Experimental outliers are circled in red.

**Figure 8 polymers-11-01912-f008:**
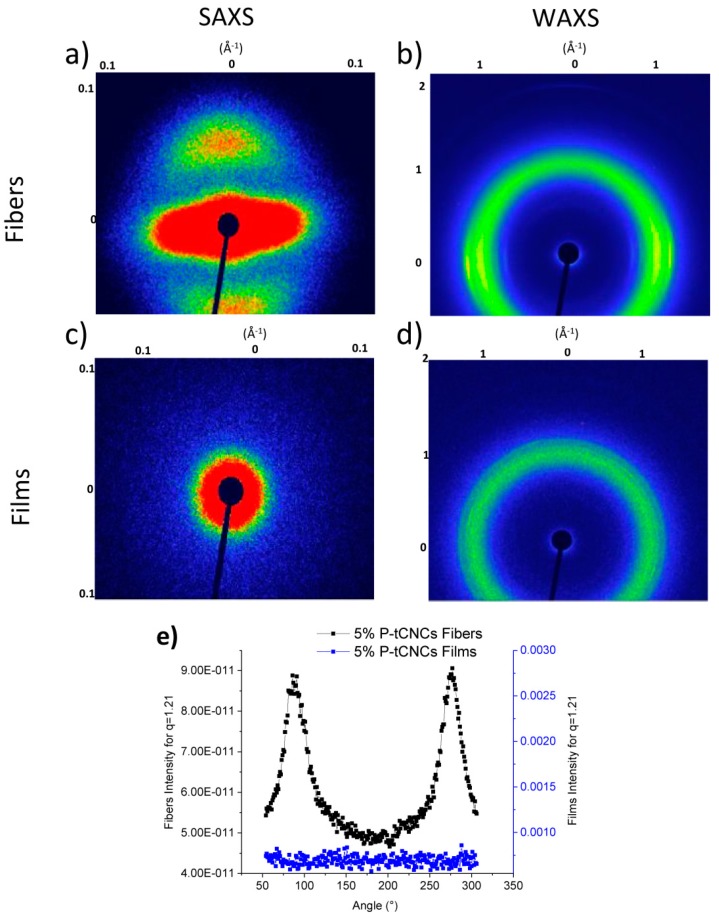
2D small and wide-angle X-ray scattering spectra of (**a**,**b**) 5 wt% P-tCNC nanocomposite fibers and (**c,d**) 5 wt% P-tCNC nanocomposite films. (**e**) Azimuthal integration of the (200) P-tCNC reflection at q = 1.21 Å^−1^.

**Figure 9 polymers-11-01912-f009:**
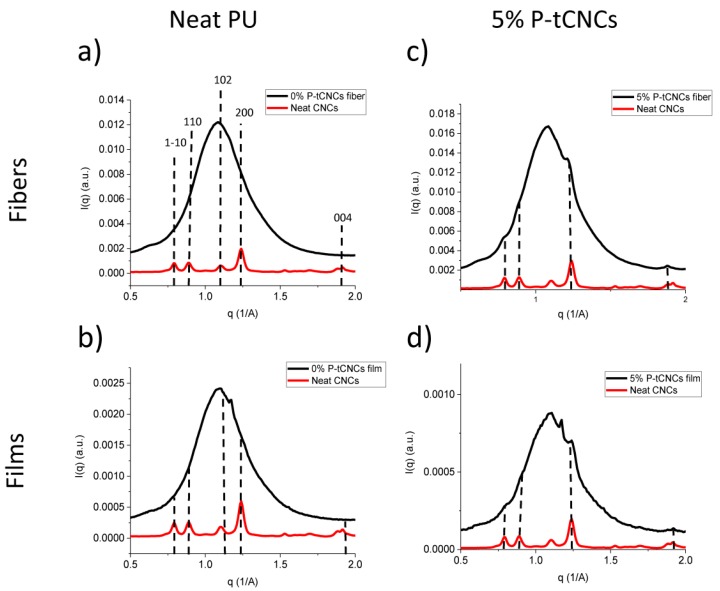
Azimuthal integration of WAXS patterns of (**a**) neat PU fibers, (**b**) neat PU films, (**c**) 5 wt% P-tCNC nanocomposite fibers, and (**d**) 5 wt% P-tCNC nanocomposite films.

**Table 1 polymers-11-01912-t001:** Young’s modulus, stress-at-break, strain-at-break, and toughness of PU and PU/P-tCNC nanocomposite fibers and films determined from a minimum of three samples.

Tensile Testing	Young’s Modulus (MPa)	Strain-at-Break (%)	Stress-at-Break (MPa)	Toughness (MJ/m^3^)
**Neat Fibers**	20 ± 3	597 ± 54	131 ± 7	2.84 ± 0.35
**1 wt% P-tCNC Fibers**	42.5 ± 5.5	422 ± 30	120 ± 12	2 ± 0.2
**5 wt% P-tCNC Fibers**	123.16 ± 10	239 ± 22	107.5 ± 4	1.59 ± 0.16
**Neat PU Films**	7.3 ± 0.01	361 ± 15	9.3 ± 0.4	0.14 ± 0.003
**1 wt% P-tCNC Films**	11.8 ± 1	340 ± 27	12.3 ± 2.5	0.19 ± 0.03
**5 wt% P-tCNC Films**	99.5 ± 2	224 ± 65	17.4 ± 2.5	0.3 ± 0.11

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
