# Peer review of "Melt-Spun Nanocomposite Fibers Reinforced with Aligned Tunicate Nanocrystals"

_polymers, 2019, doi:10.3390/polym11121912_

Round 1

Reviewer 1 Report

The authors have conducted a thorough study on the tunicate cellulose nanocrystal reinforced nanocomposite. My only concern is that research on the similar idea was published by Norma E. Marcovich, Maria L. Auad, Newton Bellesi, Steven R. Nutt, Mirta I. Aranguren titled "Cellulose micro/nanocrystals reinforced polyurethane" in 2006. Can the authors provide the insights on why this manuscript is better than the one I mentioned above?

Author Response

Comments:
The authors have conducted a thorough study on the tunicate cellulose nanocrystal reinforced nanocomposite. My only concern is that research on the similar idea was published by Norma E. Marcovich, Maria L. Auad, Newton Bellesi, Steven R. Nutt, Mirta I. Aranguren titled "Cellulose micro/nanocrystals reinforced polyurethane" in 2006. Can the authors provide the insights on why this manuscript is better than the one I mentioned above?

We thank the reviewer for appraising our work as being thorough. We also thank the reviewer for bringing the work by Marcovich et al. to our attention. We added this reference to the introduction of the revised manuscript.  However, despite similarities in the general topic, we would like to point out that focus and results of our study significantly differ from those reported by Marcovich et al. The main focus of their work is the incorporation of cellulose nanocrystals (CNCs) into polyurethane (PU) using solution-based in-situ polymerization. Our work however demonstrates melt processing of PU/CNC composites, which allowed us to fabricate nanocomposite fibers exhibiting aligned CNCs within an oriented PU matrix. Both, the melt processing of PU/CNC nanocomposites as well as the CNC alignment were not reported by Marcovich et al. Melt processing requires thermally stable CNCs, which we successfully produced via phosphoric acid hydrolysis (we confirmed thermal stability of these CNCs by TGA measurements). Note that this hydrolysis route is different from the conventional sulfuric acid hydrolysis used by Marcovich et al., which would lead to CNC degradation during thermal processing. We also note that the significantly larger aspect ratio of the tunicate CNCs used in our work compared to those employed by Marcovich et al. results in better mechanical reinforcement of polymer matrices. 

Reviewer 2 Report

The paper is good written, but can be published only after major revision. See comments below.

For both, fibers and films, you used dissolving of PU, but you used different ways to do it. Could it be better if you use the same way (with same concentration, mixing and drying regimes etc..), for example, if you would use the films (as described in section 2.5) to further extrusion? How did you control the solvent removing? Usually, for mechanical tests the constant cross-head speed (mm/min) should be used, but you used constant strain rate (100%/min). Why did you use this method? P5 L221-224: For example, sulfuric acid hydrolysis affords rather low degradation temperatures, whereas phosphoric acid hydrolysis provides high enough thermal stability for melt processing as discussed in detail by Espinosa et al. [12] for cotton CNCs and by Nicharat et al. for tunicate CNCs.[20,48]”    You have to add some information, why does the thermal stability of the CNCs depend on extraction procedure? The text on fig 2(a) is so small and barely visible, use bigger fonts, at least, as on fig 2 c Did you use a same die for all the fibers? Why does filler content greatly affect the diameter of the fibers? Fig 3: You have to give the SEM images with same magnification. Also, add low magnification images to compare fiber diameter. The inscriptions on top of the figs 7 b and c overlapped with graphs. Fig 8e:”Azimuthal integration of the spectrum in a for a q-range corresponding to the (200)” Correct, please. Can you evaluate the fibers orientation, for example, by calculating Herman’s orientation factor? Could you give any information is there segmental or macromolecular orientation in your fibers? Conclusions: “We demonstrate that the incorporation of thermally stable CNCs within a matrix of PU significantly improves the mechanical properties of the nanocomposites.” Actually, the tensile strength and elongation at break is higher for pure PU. The composites only have higher moduli. You have to add the info in conclusion sections. What is potential application of your composites? Below are some papers related to the study which can be interesting for you and can be referenced in this and your further studies:

(2019) Polymers, 11 (4), â„– 684. DOI: 10.3390/polym11040684 (2018) MATEC Web of Conferences, 242, â„– 01004. DOI: 10.1051/matecconf/201824201004 (2018) Composites Science and Technology, 164, pp. 319-326. DOI: 10.1016/j.compscitech.2018.05.032 (2017) Results in Physics, 7, pp. 1044-1045. DOI: 10.1016/j.rinp.2017.02.024 (2015) Composites Part B: Engineering, 76, pp. 79-88. DOI: 10.1016/j.compositesb.2015.02.019 (2014) Inorganic Materials: Applied Research, 5 (1), pp. 22-27. DOI: 10.1134/S2075113314010092 (2014) Inorganic Materials: Applied Research, 5 (4), pp. 386-391.DOI: 10.1134/S2075113314040194 (2014) Journal of Alloys and Compounds, 586 (SUPPL. 1), pp. S459-S463. DOI: 10.1016/j.jallcom.2012.11.048 (2014) Journal of Alloys and Compounds, 586, pp. S168-S172. DOI: 10.1016/j.jallcom.2012.12.045 (2018) Materials and Design, 156, pp. 22-31. DOI: 10.1016/j.matdes.2018.06.034 (2018) International Journal of Materials Research, 109 (8), pp. 771-778. DOI: 10.3139/146.111656 (2018) Applied Sciences (Switzerland), 8 (5), â„– 750, DOI: 10.3390/app8050750 (2018) Materials Letters, 215, pp. 288-291. DOI: 10.1016/j.matlet.2017.12.132  (2018) Micro and Nano Letters, 13 (5), pp. 588-590. DOI: 10.1049/mnl.2017.0235

Author Response

For both, fibers and films, you used dissolving of PU, but you used different ways to do it. Could it be better if you use the same way (with same concentration, mixing and drying regimes etc..), for example, if you would use the films (as described in section 2.5) to further extrusion? How did you control the solvent removing?

Preparing melt-extruded samples using the same procedure as for films is impractical. The relatively large amounts of PU required for extrusion (4 gr of PU) would have required solvent casting of 800 ml solutions of PU/CNCs in DMF, which would take prohibitively large amounts of time. We do not anticipate that this difference in processing films and fibers results in significant differences in the dispersion of CNCs with the PU matrices because in both cases the CNCs were sonicated in pure DMF prior to any further processing to ensure proper dispersion. We note that the slightly longer solvent removal times for films (3 days in an oven at 50 C, followed by 1 day in vacuum at 50 C) compared to fibers (2 days in an oven at 50 C, followed by 1 day in vacuum at 50 C) were used to ensure the complete removal of any residual solvent. This was not required for fibers, where complete solvent removal was found already after 2+1 days of drying. In both cases, solvent removal was determined based on visual inspection of the samples and their overall appearance.

Usually, for mechanical tests the constant cross-head speed (mm/min) should be used, but you used constant strain rate (100%/min). Why did you use this method?

We thank the reviewer for pointing out the unusual notation we used to describe the applied strain rate. In fact, we did use a constant cross-head speed (50 mm/min) but reported a strain of 100% /min, which was calculated using the gauge length of 50 mm. We changed this in the revised version of the manuscript.

P5 L221-224: For example, sulfuric acid hydrolysis affords rather low degradation temperatures, whereas phosphoric acid hydrolysis provides high enough thermal stability for melt processing as discussed in detail by Espinosa et al. [12] for cotton CNCs and by Nicharat et al. for tunicate CNCs.[20,48]” You have to add some information, why does the thermal stability of the CNCs depend on extraction procedure?

We are sorry to hear that the reviewer misses information on the effect of the hydrolysis procedure on the CNC thermal stability. The reason for the comparably poor thermal stability of CNCs hydrolyzed in sulfuric acid are sulfate groups grafted to the CNC surface, which catalyze their thermal degradation, as mentioned in the introduction of the manuscript (lines 79-82):
“However, melt processing of CNC-containing composites is significantly limited by the poor thermal stability of CNCs extracted by sulfuric acid-based hydrolysis. During this hydrolysis, sulfate groups are grafted and adsorbed to the surface of the CNCs, increasing their electrostatic repulsion but catalyzing their thermal degradation.”
To further clarify this point we added complementary information to the revised manuscript (lines 219-220) elaborating that phosphoric acid hydrolysis incorporates less degradation-catalyzing charged groups at the CNC and therefore renders the CNCs thermally more stable.

The text on fig 2(a) is so small and barely visible, use bigger fonts, at least, as on fig 2 c

The font size in Figure 2(a) has been increased and now matches the one in Figure 2(c).

Did you use a same die for all the fibers? Why does filler content greatly affect the diameter of the fibers?

We used the same die for all experiments. The increased diameter of fibers extruded at identical conditions is related to the increased viscosity with increasing CNC content. This is already mentioned in the manuscript (lines 269-274), and consistent with previous reports on similar systems (ref. [34]).

Fig 3: You have to give the SEM images with same magnification. Also, add low magnification images to compare fiber diameter.

All SEM images are shown on the same magnification in the revised version of the manuscript. We realize that we erroneously stated fiber diameters were determined based on SEM imaging (caption of Figure 3). However, the fiber diameters were determined based on micrometer measurements as stated in the methods. This has been corrected in the revised manuscript. We note that SEM images of the fibers were not taken at identical low magnification, and therefore not included in the manuscript.

The inscriptions on top of the figs 7 b and c overlapped with graphs.

This has been fixed. We thank the reviewer for pointing out this formatting issue.

Fig 8e:”Azimuthal integration of the spectrum in a for a q-range corresponding to the (200)” Correct, please.

This typo has been fixed.

Can you evaluate the fibers orientation, for example, by calculating Herman’s orientation factor?

The reviewer makes a very good point here. Unfortunately, the overlapping peaks of contributions coming from the PU matrix and CNC filler in both Raman spectra and WAXS profiles makes it impossible to evaluate the orientational order of the CNCs alone, and therefore does not allow the calculation of the Herman’s orientation parameter.

Could you give any information is there segmental or macromolecular orientation in your fibers?

The anisotropy in the SAXS images (Figure 8 (a)) reflects an orientation on a mesoscopic scale, i.e. several tens of nm, which we assume the reviewer refers to as macromolecular orientation. The observed anisotropy in the WAXS regime, on the other hand, reflects an orientation on the scale of several Angstroms, which is comparable the dimension of polymer segments. 

Conclusions: “We demonstrate that the incorporation of thermally stable CNCs within a matrix of PU significantly improves the mechanical properties of the nanocomposites.” Actually, the tensile strength and elongation at break is higher for pure PU. The composites only have higher moduli. You have to add the info in conclusion sections.

We thank the reviewer for pointing out the imprecise wording in the conclusion. We changed this in the revised version.

What is potential application of your composites?

Potential applications of CNC-based polyurethane nanocomposites are similar to those of other polymeric composite materials and include packaging, textile manufacture, and also medical applications to name a few. The advantage of using CNCs as reinforcement agent is their biodegradability, low cost, and great abundance as mentioned in the introduction of our manuscript.

Below are some papers related to the study which can be interesting for you and can be referenced in this and your further studies:

 (2019) Polymers, 11 (4), â„– 684. DOI: 10.3390/polym11040684 (2018) MATEC Web of Conferences, 242, â„– 01004. DOI: 10.1051/matecconf/201824201004 (2018) Composites Science and Technology, 164, pp. 319-326. DOI: 10.1016/j.compscitech.2018.05.032 (2017) Results in Physics, 7, pp. 1044-1045. DOI: 10.1016/j.rinp.2017.02.024 (2015) Composites Part B: Engineering, 76, pp. 79-88. DOI: 10.1016/j.compositesb.2015.02.019 (2014) Inorganic Materials: Applied Research, 5 (1), pp. 22-27. DOI: 10.1134/S2075113314010092 (2014) Inorganic Materials: Applied Research, 5 (4), pp. 386-391.DOI: 10.1134/S2075113314040194 (2014) Journal of Alloys and Compounds, 586 (SUPPL. 1), pp. S459-S463. DOI: 10.1016/j.jallcom.2012.11.048 (2014) Journal of Alloys and Compounds, 586, pp. S168-S172. DOI: 10.1016/j.jallcom.2012.12.045 (2018) Materials and Design, 156, pp. 22-31. DOI: 10.1016/j.matdes.2018.06.034 (2018) International Journal of Materials Research, 109 (8), pp. 771-778. DOI: 10.3139/146.111656 (2018) Applied Sciences (Switzerland), 8 (5), â„– 750, DOI: 10.3390/app8050750 (2018) Materials Letters, 215, pp. 288-291. DOI: 10.1016/j.matlet.2017.12.132  (2018) Micro and Nano Letters, 13 (5), pp. 588-590. DOI: 10.1049/mnl.2017.0235

We thank the reviewer for pointing out these papers on composites using carbon fibers and nanotubes. However, given the our manuscript’s focus on CNC-based polymer composites rather than those based on carbon fibers, which is covered extensively in the literature over several decades, we decided not to include these references in the revised manuscript.

Round 2

Reviewer 1 Report

I think we can accept the manuscript

Author Response

We thank the reviewer for reading our manuscript and his positive recommendation.

Reviewer 2 Report

«In both cases, solvent removal was determined based on visual inspection of the samples and their overall appearance.»

 It is well known that visual inspection is not the best method for materials characterization. It would be better to use IR spectroscopy (initial polymer /solvent/ produced composites) or, at least, control the mass of your samples.

«In fact, we did use a constant cross-head speed (50 mm/min) but reported a strain of 100% /min»

Keep it in mind that due to elastic/plastic behavior of materials during mechanical tests, at the same cross-head speed true strain rate of the material can be differ from the speed.

Fig 5a Y-axis: “Standard force (MPa)” Force can’t be measured in MPa. Correct it. “Unfortunately, the overlapping peaks of contributions coming from the PU matrix and CNC filler in both Raman spectra and WAXS profiles makes it impossible to evaluate the orientational order of the CNCs alone”  But according to the text:The Raman band at 1615 cm-1 373 , corresponding to the C-C ring stretching  mode of the benzene rings of the diphenylmethane motif, was used to characterize the PU orientation. … Only two P-tCNC Raman bands are observable; the one of interest at 1095 cm-1 376 , which was used for the evaluation of the orientation” you used different peaks for PU and CNC. The advantages of your composites and their potential application should be provided in conclusion section.

Author Response

In both cases, solvent removal was determined based on visual inspection of the samples and their overall appearance.» It is well known that visual inspection is not the best method for materials characterization. It would be better to use IR spectroscopy (initial polymer /solvent/ produced composites) or, at least, control the mass of your samples.

We thank the reviewer for their comment. We regret not having pointed out previously that the solvent removal protocols used in our manuscript are standard procedures in the field of CNC-based PU composites. It is commonly accepted that vacuum-drying samples at 50 °C for one day essentially removes all solvent. Nevertheless, we now include IR spectra of both fibers and films in the Supplementary Information (Figure S1) as requested by the reviewer. Based on these spectra we cannot detect any significant contribution from DMF based on a comparison to tabulated DMF IR spectra.

«In fact, we did use a constant cross-head speed (50 mm/min) but reported a strain of 100% /min». Keep it in mind that due to elastic/plastic behavior of materials during mechanical tests, at the same cross-head speed true strain rate of the material can be differ from the speed.

We thank the reviewer for this explanation.  We will take this into account in future work.

Fig 5a Y-axis: “Standard force (MPa)” Force can’t be measured in MPa. Correct it.

We thank the reviewer for pointing out this incorrect notation. We changed the axis label in the revised version of the manuscript.

“Unfortunately, the overlapping peaks of contributions coming from the PU matrix and CNC filler in both Raman spectra and WAXS profiles makes it impossible to evaluate the orientational order of the CNCs alone”  But according to the text: “The Raman band at 1615 cm-1 373 , corresponding to the C-C ring stretching  mode of the benzene rings of the diphenylmethane motif, was used to characterize the PU orientation. … Only two P-tCNC Raman bands are observable; the one of interest at 1095 cm-1 376 , which was used for the evaluation of the orientation” you used different peaks for PU and CNC.

We apologize for the lack of clarity in our first explanation. Indeed, Raman and WAXS allow for probing the orientation of CNCs and PU separately. However, their contributions can only be distinguished on a qualitative level. A quantification based on the Hermann orientation parameter (S) is impossible in the present system (CNCs/PU) because absolute values of the CNC contributions in both WAXS and Raman cannot be obtained due to overlapping shoulders of peaks corresponding to CNC and PU. To give a further explanation, the tails of the PU peak at 1120 cm-1 clearly overlap with the CNC peak at 1095 cm-1 as shown in the inset of Figure 7a. This overlap of the tails of the PU peak with the CNC peak make it impossible to decouple the contributions stemming from CNC and PU orientation, which is critical to obtain the baseline intensity of CNC peaks, which in turn are required for the calculation of the Hermann orientation factor.

For WAXS, a similar problem occurred, as shown in Figures 9c and 9d: Here, the weak CNC scattering peaks (due to the low CNC content) sit on an intense and broad PU reflection, which entirely obscures the CNC contribution at angles around 180°, which is essential for determining reliable “baselines” for the Hermann orientation factor.

In other words, while peak positions of CNC and PU contributions can be distinguished in WAXS and Raman, this is not the case for the absolute intensities.

Therefore, we deliberately decided to only qualitatively evaluate the orientation of the nanocomposite fibers and films in the present manuscript.

The advantages of your composites and their potential application should be provided in the conclusion section.

The system (PU/CNC) used is a well-known and well-researched polymer composite (see literature overview provided in the manuscript). Broad applications of this system could be imagined but discussing those is beyond the scope of the present manuscript. Here our goal is to focus on the fundamental understanding of the effect of the orientation on the mechanical properties.

Round 3

Reviewer 2 Report

Best wishes!